# Thermal Inactivation of Hepatitis E Virus in Pork Products Estimated with a Semiquantitative Infectivity Assay

**DOI:** 10.3390/microorganisms11102451

**Published:** 2023-09-29

**Authors:** Melissa Stunnenberg, Suzanne C. van Huizen, Arno Swart, Willemijn J. Lodder, Ingeborg L. A. Boxman, Saskia A. Rutjes

**Affiliations:** 1Laboratory for Zoonosis and Environmental Microbiology, Centre for Infectious Disease Control, National Institute for Public Health and the Environment (RIVM), P.O. Box 1, 3720 BA Bilthoven, The Netherlands; melissa.stunneberg@rivm.nl (M.S.); suzanne.van.huizen@rivm.nl (S.C.v.H.); arno.swart@rivm.nl (A.S.); willemijn.lodder@rivm.nl (W.J.L.); 2National Reference Laboratory Food-Borne Viruses, Wageningen Food Safety Research (WFSR), Wageningen University and Research, P.O. Box 230, 6700 AE Wageningen, The Netherlands; ingeborg.boxman@wur.nl

**Keywords:** HEV infectivity, food-borne, HEV-3c, HEV-3e, cell culture method, pork matrices, immunofluorescence, food processing

## Abstract

Hepatitis E virus genotype 3 (HEV-3) is a food-borne pathogen causative of hepatitis E infections in humans. In Europe, HEV-3 is mainly transmitted through the consumption of raw or undercooked pork. In order to determine the effectiveness of control measures that can be taken in the industry or by the consumer, it is pivotal to determine the infectivity of HEV present in pork products after thermal food-processing steps. First, we implemented a method for the detection of infectious HEV-3c and HEV-3e in a cell culture medium and in extracts from inoculated pork products. Next, we investigated the effect of the thermal inactivation of HEV by mimicking food-processing steps specific for dried sausage and liver homogenate matrices. After four weeks, HEV-inoculated dried sausage subjected to 21 °C or lower temperatures was still infectious. For the liver homogenate, the highest HEV-3c/e inactivation of the conditions tested was observed at 71 °C for five min or longer. Finally, our method was able to successfully detect and estimate viral loads of infectious HEV in naturally infected pig livers. Our data provide a basis for the future use of the quantitative microbial risk assessment of infectious HEV in pork products that are subjected to thermal food processing steps.

## 1. Introduction

Hepatitis E virus (HEV) is the causative agent of hepatitis E in humans. The zoonotic HEV genotypes 3 (HEV-3) and 4 (HEV-4) are predominantly responsible for hepatitis E cases in Europe [1,2]. Over the past decade, the number of autochthonous hepatitis E cases in humans has increased in Europe, including the Netherlands [3,4]. Although the direct cause of the observed rise in HEV-infected individuals is still unknown, the similarities between HEV-3 strains identified in humans and animals suggest that animal reservoirs harboring HEV-3 play a role in HEV-3 infections in humans [5].

Domestic pigs have been described to be the main reservoir for HEV, with HEV found in pigs that were raised at conventional, free-range and organic farms and also in different breeding systems [6,7,8,9]. The dominant HEV-3 subtype in pigs in the Netherlands is 3c; however, the European autochthonous subtypes 3e and 3f have also been detected [6,10]. Neither living in close proximity to pig farms nor direct contact with HEV-positive pigs are considered to be important HEV transmission routes with regard to public health [11,12]. In contrast, contact with HEV-containing (waste) water and the consumption of dried sausages composed of raw pork are important risk factors for HEV infection, of which the latter one is considered the primary route of zoonotic HEV transmission to humans [11,13,14]. In the Netherlands, HEV RNA has been detected in ready-to-eat raw meat sausages, pig liver and blood products, and was subtyped as HEV-3c [15,16,17,18]. Notably, it has been shown that HEV-3 sequences recovered from raw pig liver sausage were genetically linked to sequences recovered from patients who consumed pig liver sausage [19].

Pork and liver are often subjected to different industrial food-processing techniques, such as fermentation, acidification, high-pressure processing steps, or to thermal treatments, of which the latter one may also be performed by the consumer. During meat processing, both meat and meat products are subjected to a wide variety of different temperatures for different durations [20]. For example, the preparation of ragout is performed by boiling, where the core temperature of the meat reaches at least 80 °C, whereas sausages (e.g., liverwurst) are heated with temperatures of 75–80 °C [20]. The drying of meat (e.g., raw pig sausages) occurs at low temperatures [20]. Finally, the consumer stores most meat or meat products in the refrigerator or freezer. Ready-to-eat meats might be consumed without further preparation, while other products, such as tenderloin or hams, are generally heated to ±50–60 °C [21,22].

Different experimental studies have evaluated the effect of thermal conditions on the presence of infectious HEV, and their findings are visualized in Appendix A. Studies have either obtained presence/absence data [23,24] or quantitative data [25,26,27] of HEV-3 strains. Imagawa et al. detected HEV RNA after subjecting a cell culture medium and minced pork to different temperatures, ranging from 56 to 80 °C, for 1–60 min [24]. Feagins et al. inoculated pigs with heat-treated commercial pig livers (56 °C for 1 h and to 191 °C for 5 min) and, subsequently, analyzed fecal and serum samples for the presence of HEV RNA as a read-out for remaining HEV infectivity in the inocula [23] (Appendix A). Quantitative studies on cell culture media and two different liver matrices (food products prepared with HEV-3-contaminated liver and wild boar liver suspension) were performed to assess the remaining HEV RNA as a measure for residual HEV and, thus, indirectly, for the quantification of HEV inactivation (Appendix A) [25,27]. The quantification of infectious HEV in a cell culture medium treated for 1 min at temperatures ranging from 65 to 80 °C was consistent with the HEV RNA presence/absence data (Appendix A) and, importantly, revealed a more detailed inactivation pattern (Appendix A). HEV-containing liver matrices were either exposed to temperatures of up to 37 °C [27] or to 62 °C or more [25]. However, quantitative studies on the heating of pork products at lower temperatures (±55 °C) (e.g., for pork tenderloin) or similar temperatures for different durations have not yet been performed.

Detection techniques that are able to discriminate between an infectious and inactivated virus are needed in order to correlate the presence of HEV RNA in pork products with a public health risk, which standard molecular RNA detection techniques are incapable of. The thermal stability of infectious HEV has been studied in cell culture media [26] (Appendix A); however, the extent of thermal HEV inactivation within pork products using culture methods has, to our knowledge, not yet been described. Collectively, these studies highlight gaps in the literature, which we attempted to address in our study.

Here, we first implemented a cell culture method combined with an immunofluorescence (IF) technique to demonstrate the replication of HEV-3c and HEV-3e. Next, the method was optimized for application on viral extracts from inoculated dried sausage and liver. For this, methods were developed to extract the virus from these matrices while maintaining viral infectivity. With this experimental set-up, the fate of HEV-3c/e infectivity in sausage and liver matrices was assessed in a variety of experiments that mimicked food-processing temperatures as performed by the industry or the consumer. These data were used to estimate HEV-3c/e inactivation through Bayesian MPN modelling. Finally, our method was applied to naturally infected pig livers that were previously shown to be HEV-RNA-positive. Taken together, assessing the effect of food-processing temperatures on the infectivity of HEV in pork products is important to evaluate the risk of infection through the consumption of HEV-contaminated pork products.

## 2. Materials and Methods

### 2.1. Virus Stock Preparation

HEV strains 14-16753 (HEV-3c) and 14-22707 (HEV-3e), isolated from HEV patients and confirmed with sequence analyses, were used for propagation in the human liver carcinoma cell line PLC/PRF/5, as previously described [28]. Both isolates were provided by Dr. Mathias Schemmerer (Institute of Clinical Microbiology and Hygiene, University Medical Center Regensburg, 93053 Regensburg, Germany). In brief, PLC/PRF/5 cells were seeded (approximately 2.0 × 10^3^ cells/mL in a T75 flask) in Dulbecco’s modified eagle medium (DMEM) with 10% fetal bovine serum (FBS, 10270106, Gibco, Waltham, MA, USA) and 100 µg/mL gentamicin (15750060, Gibco, Waltham, MA, USA) at least five days prior to inoculation. When the cells were at an approximate 95% confluency, the medium was replaced with 30 mL DMEM containing 1.5 mL (defrosted) HEV-3c or HEV-3e with an unknown titer, for which the number of viral particles has not yet been estimated. This was supplemented with 2% FBS (10270106, Gibco), 50 µg/mL gentamicin (15750060, Gibco, Waltham, MA, USA), 25 µM deoxycholic acid (DCA, D2510, Merck—Millipore, Burlington, MA, USA) and 200 µL 100× of the antibiotic antimycotic (AA, A5955, Merck—Millipore, Burlingotn, MA, USA), from now on referred to as complete DMEM. After seven days, the medium was replaced with complete DMEM (without HEV) and incubated for another seven days. At day 14, the culture medium was collected (first collection). New complete DMEM (12 mL) was added to the cells. The cells were then lysed using two freeze–thaw cycles (−80 °C). The collected suspension was centrifugated at 175× *g* for 10 min to separate the cell fragments from the released virus present in the supernatant. The supernatant derived from the lysed cells was pooled with supernatant from the first collection and stored at −80 °C (HEV-3c and HEV-3e stocks). HEV-3c experiments were performed with one stock. A total of two stocks were prepared of HEV-3e, and these were used separately (not pooled) for different experiments. All cell cultures were kept at 37 °C with 5% CO_2_.

### 2.2. HEV Extraction from Pork Matrices and Naturally Infected Pig Livers

Subsamples (350–400 mg) were produced from dried raw pig sausage, labeled as “harde boeren borrelworst”, and liver. Both originated from Dutch-reared pigs and obtained from a local butcher. A total of two dried sausages and two livers were used to perform the experiments. All noninoculated products were negative for HEV RNA, as tested using the quantitative reverse transcription polymerase chain reaction (RT-qPCR, Section 2.6). Dried sausage was cut into small pieces of approximately 1 mm by 1 mm, while the liver was homogenized (from now on referred to as liver homogenate) using a laboratory blender (Waring, Z27221, Merk—Millipore, Burlington, MA, USA). Pork matrices were inoculated with 500 µL HEV-3c or HEV-3e stock. Inoculated samples were, subsequently, exposed to a series of thermal treatments, mimicking food-processing steps (Section 2.3). After the treatments, 500 µL PBS was added to the inoculated pork matrix. Subsequently, HEV was extracted from the matrix by mixing the products in microcentrifuge tubes with 2.0 mm diameter yttria-stabilized zirconium oxide beads (Lysing Matrix Z, 6961100, MP Biomedicals, Irvine, CA, USA) to disrupt the tissue integrity using a FastPrep machine (Savant Bio 101 FastPrep FP120 Cell disruptiesysteem, 05298, Gemini, Apeldoorn, The Netherlands) at 4.0 m/s for 30 s. Liver homogenates were mixed once, while the dried sausage samples were mixed three times using the FastPrep machine with a 2 min waiting step to prevent overheating. Then, the pork product suspensions were centrifugated at 10,000× *g* at room temperature for 1 min. The supernatants were collected in a clean centrifuge tube. For the dried sausage, the extract volume varied between 400 and 700 µL. The extract volume for the liver homogenate varied between 800 and 1100 µL. To remove coextracted residual fat and to prevent the growth of yeast, 200 µL chloroform/mL pork extract was added. The samples were vortexed at 2700 rpm for 15 s followed by incubation at room temperature for 15 min. Samples were then centrifuged at 10,000× *g* at 4 °C for 15 min and the supernatant from the upper layer (water phase) was collected in a clean tube. The samples were filtered using Millex PVDF syringe 0.2 µm filters (Merck—Millipore, Burlington, MA, USA). After filtering, the viral extract volumes for the dried sausage varied between 250 and 400 µL, while the viral extract volumes for the liver homogenate varied between 400 and 600 µL. The expected volumes depended mainly on the fat content of the product and the subsequent homogenization efficiency. Viral extracts were also prepared from ten livers originating from Dutch slaughterhouse pigs that tested HEV-positive using RT-qPCR in 2019 [17]. In addition, four other liver samples were extracted that were from more recent monitoring studies (2020 (n = 3) and 2023 (n = 1)) carried out for the Netherlands Food Safety and Consumer Authority (NVWA) by Wageningen Food Safety Research (WFSR). The typing of these livers [15] revealed the presence of the HEV-3c subtype [17]. One of the selected livers served as a matrix control (blank) and tested negative for HEV RNA. Previous liver RT-qPCR results had not been shared with the National Institute for Public Health and the Environment (RIVM) prior to executing the immunofluorescence (IF) experiments (Section 2.5). To obtain the liver homogenates, liver subsamples of ±400 mg in 1 mL PBS were processed using the microcentrifuge tubes with 2.0 mm diameter yttria-stabilized zirconium oxide beads (Lysing Matrix Z, 6961100, MP Biomedicals, Irvine, CA, USA) with the FastPrep machine (two to three times, with 2 min waiting steps in between cycles). Then, subsequent extractions were performed in a similar manner to that performed for the spiked pork products. The volume of the viral extracts varied between 400 and 550 µL. For testing whether liver extracts were positive for HEV RNA, 200 µL of naturally infected liver extracts was used for the RNA extraction, eluted in 50 µL and 5 µL and tested with RT-qPCR (Section 2.6).

### 2.3. Food Processing Conditions

The remaining HEV infectivity in the medium (500 µL) and inoculated pork products after exposure to a wide variety of time–temperature combinations was assessed. The selected conditions (Table 1) were chosen to be complementary to the existing data from the literature, as visualized in Appendix A. A refrigerator (4 °C), incubators (10, 15, 21 °C) and water baths (55, 65, 71, 80 °C) were used to achieve stable and controlled temperatures. The water baths were controlled with the use of internal and external thermometers. The experiments were performed on different days (biological replicates, Table 1), with a minimum of four different wells of A549/D3 cells inoculated per dilution (experimental replicates, Section 2.4). As explained in the Result Section 3.1, combined HEV-3c and HEV-3e data were used as input data for the Bayesian MPN model, resulting in the estimation of combined HEV-3 (HEV-3c/e) inactivation rates based on 2 or 3 biological replicates.

### 2.4. Cell Culture Method

A cell culture method for the assessment of HEV infectivity was implemented to detect infectious HEV-3c and HEV-3e. For this method, the human lung carcinoma subclonal cell line A549/D3 [29], which was kindly provided by Dr. Reimar Johne (Department of Biological Safety, German Federal Institute of Risk Assessment, 10589 Berlin, Germany), was used. The cells were seeded (1.2 × 10^4^ cells/mL) in a 96-well flat-bottom plate in minimum essential medium (MEM) with 10% heat-inactivated FBS and 100 µg/mL gentamicin, and grown at a ±95% confluency. At least five days after seeding, the cells were inoculated with 25 µL HEV-3c or HEV-3e (stock, in medium, extracted from meat matrices or extracted from naturally infected pig livers (Section 2.2)) in 125 µL complete MEM (MEM supplemented with 5% FBS, 50 µg/mL gentamicin, 25 µM DCA and 11 µL 100× AA). In order to estimate the concentration of infectious HEV with the Bayesian most probable number (MPN) model (Section 2.7), a minimum of four different wells of A549/D3 cells were inoculated per dilution. HEV in the medium was both not diluted and diluted 5, 10, 100, 10^3^,10^4^, 10^5^ and 10^6^ times. The inoculation of A549/D3 cells with dried sausage and liver homogenate was performed similarly, although inoculation with undiluted pork products was not performed. A549/D3 cells were inoculated with suspensions prepared from naturally infected pig livers. A total of 14 livers were tested. Per liver, eight wells of A549/D3 cells were inoculated per dilution (5, 10 and 100 times). After seven days, the medium was replaced with complete MEM (without HEV inoculate) and incubated for another seven days. After incubation, the cells were assessed for the presence of viable, infectious HEV using IF (Section 2.5). All cell cultures were kept at 37 °C with 5% CO_2_.

### 2.5. Immunofluorescence Detection Assay

Intracellular HEV capsid protein staining was used as a measure of HEV replication and visualized using IF. A549/D3 cells were fixed with 4% paraformaldehyde in PBS, incubated at room temperature for 10 min and, subsequently, washed with PBS. The cells were either directly used for IF or stored at 4 °C in the dark. For IF, cells were permeabilized with 0.2% triton X-100 in PBS at room temperature for 10 min. Then, the cells were washed twice with PBS, followed by incubation with 1% FBS in PBS as the blocking agent at 37 °C for 30 min. After the removal of the blocking agent, the cells were treated with polyclonal rabbit anti-HEV antibodies against the HEV capsid protein [30] (antiserum 8282, kindly provided by Dr. Rainer Ulrich, Friedrich-Loeffler-Institute, Federal Research Institute for Animal Health, 17493 Greifswald-Insel Riems, Germany) using a 1:500 dilution in 1% FBS in PBS at 37 °C for 30 min. Then, the cells were washed three times with PBS, followed by incubation with fluorescein isothiocyanate (FITC)-conjugated goat antirabbit antibodies (F9887, Sigma-Aldrich, St. Louis, MI, USA) in a 1:1000 dilution in 1% FBS in PBS at 37 °C for 30 min and protected from light. The cells were washed twice with PBS and once with milli-Q. The cell nucleus was visualized using a 4′,6-diamidino-2-phenylindole (DAPI) stain (HP20.1, Carl Roth, Karlsruh, Germany). The cells were protected from light and kept at 4 °C until examination with a fluorescence microscope (DMi8, Leica microsystems, Wetzlar, Germany). For the Bayesian MPN model (Section 2.7), a ‘positive’ call was assigned to every well containing cytoplasm-localized green fluorescence (both low and high numbers of fluorescent positive cells were scored as ‘positive’). Wells without cell-localized fluorescence were assigned as ‘negative’. For each sample, the dilution at which no fluorescent cells were observed in any of the replicate wells was determined.

### 2.6. Reverse Transcription Polymerase Chain Reaction

RNA extraction was performed using the RNA Direct-zol^TM^ miniprep (Zymo Research, Irvine, CA, USA), according to manufacturer’s instructions. In brief, 150 µL of processed dried sausage or liver homogenate or 200 µL extract from naturally infected pig livers was used for RNA extraction. The DNAse treatment was performed with 5 µL DNAse I (6 U/µL) at room temperature for 15 min. The samples were eluted in 50 µL, and 5 µL per sample was tested with RT-qPCR. Equine arteritis virus (EAV) was used as an internal control [31]. The following reagents were used per RT-qPCR reaction: 1x TaqMan Fast Virus 1-step Master Mix (Invitrogen, Waltham, MA, USA), 0.5 µM forward primer, 0.5 µM reverse primer, 0.25 µM probe, RT-PCR-grade water (Sigma-Aldrich, St. Louis, MI, USA), 5 µL RNA, in a total reaction volume of 20 µL. Oligonucleotides were used for the amplification of *ORF2/3*. The following primers and probe were used [32]: forward primer: 5′-GGTGGTTTCTGGGGTGAC-3′; reverse primer: 5′-AGGGGTTGGTTGGATGAA-3; probe: FAM-TGATTCTCAGCCCTTCGC-BHQ1. The reverse transcription and subsequent DNA amplification were performed on a LightCycler 480 (Roche, Basel, Switzerland) using the following cycle settings: 50 °C for 30 min (4.4 C/s), 95 °C for 5 min (4.4 C/s), amplification stage (50 cycles) of two steps: 95 °C for 10 s (4.4 C/s), followed by 55 °C for 30 s (2.2 C/s), 37 °C for 30 s (2.2 C/s). The screening of the livers for HEV was performed at WFSR, as previously described [15,33].

### 2.7. Bayesian MPN Model

Scoring results of the green fluorescent cells after IF were used to parameterize a model, which could be considered as a variation of the ‘most probable number’ (MPN) method in a Bayesian framework. We did not aim to build a model that determined the time and temperature dependence, but, rather, calculated the inactivation for each time–temperature combination. The model describes the data-generating process: starting with an unknown virus stock concentration and the probabilities of virus particles (1) surviving the inactivation process and (2) being detected at a certain dilution. The model was stratified with the HEV-3 subtype (3c/3e) and matrix (no matrix (=medium)/dried sausage/liver homogenate). Starting from priors for the unknown parameters (stock concentration, inactivation parameters), the observed data and model description were combined to yield posterior estimates, reflecting the uncertainty in our estimate. Model fitting was performed using Stan [34], interfaced from R v4.3.0 [35].

#### 2.7.1. Bayesian MPN Model for Medium, Dried Sausage and Liver Homogenate

Experiments were performed with two different subtypes of HEV (HEV-3c and HEV-3e), three matrices and two different stocks for HEV-3e. These conditions were taken into account when building the model. For data point i, we defined the following variables:$$s_{i}=\left\{\begin{array}{cc} 1 & \mathrm{for}\;\mathrm{subtype}\;3e\\ 2 & \mathrm{for}\;\mathrm{subtype}\;3c \end{array}\right.$$
$$m_{i}=\left\{\begin{array}{cc} 1 & \mathrm{for}\;\mathrm{matrix}\;‘\mathrm{medium}’\\ 2 & \mathrm{for}\;\mathrm{matrix}\;‘\mathrm{dried}\;\mathrm{sausage}’\\ 3 & \mathrm{for}\;\mathrm{matrix}\;‘\mathrm{liver}’ \end{array}\right.$$
$$b_{i}=\left\{\begin{array}{cc} 1 & \mathrm{for}\;\mathrm{stock}\;1\\ 2 & \mathrm{for}\;\mathrm{stock}\;2 \end{array}\right.$$

Furthermore, a range of times $\left(t_{i}\right)$ and temperatures $\left(T_{i}\right)$, which were also used as categorical variables (starting the count at one), were used. Finally, we defined a variable $e_{i}$ related to the experiment, which had a unique label depending on the combination of matrix, time and temperature.

The concentration after the preparation of the matrix was denoted as $C_{0,i}\left(s_{i},\;m_{i},\;b_{i}\right)$ in particles per mL. Since 0.5 mL of stock was considered in a single experiment, together with 0.5 mL of PBS, the concentration of particles in the 1 mL sample was:$$C_{1,i}=0.5\times C_{0,i}\left(s_{i},\;m_{i},\;b_{i}\right)$$

Note that for $m_{i}=1$ denoting the matrix ‘medium’, we directly looked at the stock, and $C_{1,i}\left(s_{i},\;1,\;b_{i}\right)$ was interpreted as the stock concentration. The concentrations for $m_{i}=2,3$ also included the matrix inhibitory effects on the concentration.

An amount of $r_{i}$ mL was recovered. We corrected for this concentration effect:$$C_{2,i}=C_{0,i}\left(s_{i},\;m_{i},\;b_{i}\right)\times 0.5/r_{i}$$

For experiments with stock only and no matrix (i.e., $m_{i}=1$, the matrix ‘medium’), there was no recovery needed, and the 0.5 mL PBS of the first step was not used. We set $r_{i}=0.5$ for the medium, so that $C_{2,i}=C_{0,i}$ in this case.

The expected number of particles after dilution with a factor $d_{i}$ was simply a division:$$C_{2,i}=C_{0,i}\left(s_{i},\;m_{i},\;b_{i}\right)\times 0.5/\left(r_{i}\times d_{i}\right)$$

We modelled the effect of heat inactivation by a factor $\gamma \left(e_{i},\;s_{i}\right)$ dependent on the experiment ($e_{i})$ and subtype ($s_{i}$). These effects were modelled additively:$$\gamma \left(e_{i},\;s_{i}\right)=\gamma_{\mathrm{experiment}}\;\left(e_{i}\right)+\gamma_{\mathrm{subtype}}\left(s_{i}\right)\;$$
$$C_{3,i}=\gamma \left(e_{i},\;s_{i}\right)\;C_{0,i}\left(s_{i},\;m_{i},\;b_{i}\right)\times 0.5/\left(r_{i}\times d_{i}\right)$$

For the experiments with no heating applied (0 s), we fixed γ to 1. Finally, 0.025 mL of the sample was examined in one well, and the expression for the average number of particles became:$$N_{i}=\gamma \left(e_{i},\;s_{i}\right)\;C_{0,i}\left(s_{i},\;m_{i},\;b_{i}\right)\times 0.5\times 0.025/\left(r_{i}\times d_{i}\right)$$

The number of particles after inactivation and observed at a certain dilution was then Poisson-distributed:$$M_{i}\sim \;\mathrm{Poisson}\left(N_{i}\right)$$

We placed uninformative priors on each of the parameters to be estimated:$$\log_{10}\left(C_{0,i}\left(s_{i},\;m_{i},\;b_{i}\right)\right)\sim N\left(4,4\right)$$
$$\log_{10}\left(\gamma_{\mathrm{subtype}}\left(1\right)\right)=0$$
$$\log_{10}\left(\gamma_{\mathrm{subtype}}\left(2\right)\right)\sim N\left(0,2\right)$$
$$\log_{10}\left(\gamma_{\mathrm{experiment}}\;\left(e_{i}\right)\right)=0,\;\mathrm{for}\;i\;\mathrm{with}\;t_{i}=0$$
$$\log_{10}\left(\gamma_{\mathrm{experiment}}\;\left(e_{i}\right)\right)\sim N\left(0,5\right),\;\mathrm{for}\;i\;\mathrm{with}\;t_{i}>0$$

Next, we defined the likelihood function. The outcome of experiment i was $y_{i}$, which was either 0 or 1, depending on a positive result being observed or not, respectively. Since the distribution of particles $M_{i}$ was Poisson, the probability of observing more than 0 was:$$P\left(M_{i}>0\right)=1-\exp(-N_{i})$$

An outcome of $y_{i}$ was the result of a Bernoulli draw:$$y_{i}\sim \mathrm{Bernoulli}(1-\exp(-N_{i}))$$

This likelihood, together with the priors, constituted the data analysis model. Data were representative of 2 or 3 experiments performed on different days depending on the time–temperature combination (Table 1). Model convergence was assessed by considering the stability of the trace plots of a few selected parameter values (Appendix A) [36].

#### 2.7.2. Bayesian MPN Model for Naturally Infected Pig Livers

We added 1 mL of PBS to 0.4 g of naturally infected pig liver. The concentration of HEV in the livers was C particles per gram. Then, before the sample preparation, we had the average particles in 0.4 g liver plus 1 mL PBS of:$$N_{0}=0.4C$$

After the sample preparation, there was a yield of Y mL. This contained the same number $N_{0}$ of particles. Hence, the concentration was, on average:$$C_{0}=N_{0}/Y=0.4C/Y$$

We had the dilution factors D=1,5,10,…. The concentration had to be divided by this factor:$$C_{1}=\frac{0.4C}{YD}$$

Finally, of this concentration, 0.025 mL was examined for the absence/presence of IF. In this volume, we had an expected number of particles:$$N=0.025\frac{0.4C}{YD}$$

The distribution around this rate was Poisson:$$N∼\mathrm{Poisson}\left(0.025\frac{0.4C}{YD}\right)$$

This implied that the probability of a nonzero outcome was:$$P_{\mathrm{presence}}=1-\exp\left(-N\right)$$

The observed outcome *X* of 0 or 1 particles was Bernoulli-distributed:$$X\sim \mathrm{Bernoulli}\left(P_{\mathrm{presence}}\right)$$

#### 2.7.3. Region of Practical Equivalence (ROPE)

In Bayesian statistics, the region of practical equivalence (i.e., the ROPE) is a method of assessing the relevance of a difference between parameters [37]. In brief, the region is to be defined by the researcher and indicates the interval in which a difference between the parameters is judged to be too small to be meaningful. We chose 0.5 as a meaningful difference between inactivation factors (i.e., a ROPE interval of (−0.5, 0.5)) based on the typical uncertainty range found for the stock concentrations (Appendix A). The 89% highest density interval (HDI) of the distribution of the difference was compared to the ROPE, and the following outcomes were defined:The HDI was contained within the ROPE: there was no meaningful difference;The HDI was completely outside the ROPE: there was a meaningful difference;The HDI overlapped the ROPE: we withheld judgment.

In contrast to the classical *p*-value, which is a measure of effect occurrence, the aforementioned statements were direct quantifiers of effect size, i.e., providing information on the value of the difference between the parameters of interest. Moreover, option 3 (unclear if the difference was meaningful) ultimately changed into 1 or 2 when sufficient data were present, while the *p*-value classified even the smallest difference as significant, given enough data.

## 3. Results

### 3.1. Determining Infectious HEV in Cell Culture Medium Subjected to Thermal Food-Processing Steps

Here, we set out to detect both HEV-3c and HEV-3e in our infectivity assays. The presence of infectious, replicated HEV-3c or HEV-3e in the medium was detected using the A549/D3 cell culture method, followed by the detection of HEV capsid protein in the cytoplasm (Appendix A). The dilution of the HEV inoculate (5, 10 and 100×) led to a decrease in infectious HEV, as shown by a decrease in the number of green fluorescent cells (Appendix A). The wells with cytoplasm-localized green fluorescent signals were scored and used to estimate the stock concentrations of both HEV subtypes using Bayesian MPN modelling and under the assumption that full virus recovery (no loss of viral particles) occurred (Appendix A and Appendix A). The HEV-3c inoculates reached a concentration of 6.0 log_10_ viral particles/mL, while the HEV-3e inoculates reached a concentration of 4.3 log_10_ viral particles/mL and 6.1 log_10_ viral particles/mL, depending on the stock that was used (Appendix A).

Next, we subjected HEV-3c and HEV-3e, in parallel experiments and in the absence of a matrix (i.e., cell culture medium), to a variety of thermal conditions (Table 1). The conditions were selected to complement the existing literature on HEV inactivation studies and mimic food-processing steps (Appendix A). First, the difference in inactivation factors between the HEV-3 subtypes was assessed. To this end, the difference between HEV-3c and HEV-3e inactivation factors was estimated (parameter $\gamma_{\mathrm{subtype}}$ in the model, Section 2.7.1, Appendix A). The difference in inactivation between the HEV-3 subtypes could be seen in the figure to be ±0.25 log_10_, while the typical uncertainty range in estimating the number of viral particles/mL (±0.5 log_10_, Appendix A) was larger. This provided a rationale for combining HEV-3c and HEV-3e data as input data for our Bayesian MPN model and, thus, for the estimation of combined HEV-3 (from here on referred to as HEV-3c/e) inactivation in the cell culture medium.

We next subjected HEV-3c and HEV-3e to different thermal treatments. The inactivation of the different HEV-3 subtypes and stocks in the absence of the thermal treatment was defined as 0 log_10_ inactivation at 0 weeks (Figure 1). Subjecting HEV-3c/e in the cell culture medium to 4, 10 and 21 °C for 1 week showed a slight reduction in infectious HEV (Figure 1 and Appendix A). The subjection of HEV-3c/e to 21 °C for 2 weeks resulted in a meaningful level of HEV-3c/e inactivation, as estimated by comparing the 89% highest density interval (HDI) of the distribution of the difference in HEV-3c/e inactivation to our set region of practical equivalence (ROPE) interval (Section 2.7.3). At 4 weeks of incubation at 21 °C, a 3.29 log_10_ reduction in infectious HEV-3c/e compared to the untreated inoculum was observed. In contrast to 21 °C, the subjection of HEV-3c/e to 4 and 10 °C for 1–4 weeks did not result in any meaningful levels of HEV inactivation.

HEV-3c/e in the cell culture medium was also exposed to temperatures as high as 65, 71 and 80 °C for short durations of 10 or 20 min only (Figure 1, Table 1). Subjection to 65 °C for 10 and 20 min led to a meaningful reduction in infectious HEV-3c/e (1.86 log_10_ and 3.62 log_10_, respectively). Incubation at 71 or 80 °C led to over 6.0 log_10_ HEV-3e inactivation already after 10 min of the heat treatment, suggesting that no infectious HEV remained (Figure 1 and Appendix A).

### 3.2. Determining Infectious HEV in Pork Matrices Subjected to Thermal Food-Processing Steps

We set out to test the remaining HEV infectivity in viral extracts from matrices that had been subjected to different food-processing temperatures. First, chopped dried raw pig sausage and liver homogenate were inoculated with HEV-3c or HEV-3e. After inoculation, extracts were created and infectious HEV-3c/e concentrations were estimated using the Bayesian MPN method. The HEV extracts from dried sausage contained an estimated concentration of 5.1 log_10_ (HEV-3c) and 2.7 and 4.7 log_10_ (HEV-3e) viral particles/mL. Of note, concentrations after extraction from the liver homogenate were estimated at 4.7 log_10_ (HEV-3c) and 4.1 and 5.0 log_10_ (HEV-3e) viral particles/mL (Appendix A). Although these data suggested that the virus was either lost or inactivated during the virus extraction, the viral extracts from the HEV-3c- and 3e-inoculated meat matrices still allowed for the detection of infectious HEV-3c and 3e.

Next, the meat matrices were subjected to different food-processing temperatures, specific for the matrix concerned, and HEV inactivation was assessed (Table 1). Subjecting HEV-3c/e in dried sausage to 4, 10, 15 and 21 °C for 1–3 weeks showed a minimal reduction in infectious HEV (Figure 2 and Appendix A), while subjection to 4, 10 and 15 °C for 4 weeks led to meaningful reductions in infectious HEV-3c/e of 1.20, 0.86 and 1.45 log_10_, respectively. Of note, the reductions in infectious HEV-3c/e observed at 4 and 10 °C for 4 weeks were higher than those observed in the absence of a matrix (Figure 1 and Appendix A). Subjection to 21 °C for 4 weeks led to more HEV-3c/e inactivation (2.35 log_10_) than subjection to 4, 10 and 15 °C for 4 weeks. However, remaining infectious virus was still detected (Figure 2 and Appendix A). A meaningful reduction in infectious HEV-3c/e was observed after 2 weeks of incubation at 21 °C. This was also the case in the absence of a matrix (Figure 1 and Figure 2; Appendix A).

The liver homogenates were subjected to higher temperatures than the dried sausages (Table 1). The subjection of the HEV-3c/e-containing liver homogenate to 55 °C for 10 min and to 65 °C for 5 min led to a meaningful difference in HEV-3c/e inactivation (1.31 and 1.99 log_10_, respectively). Furthermore, subjection to 55 and 65 °C for 60 min led to 1.98 and 3.56 log_10_ inactivation, respectively, and prolonging the incubation at 55 °C to 120 min did not further enhance HEV-3c/e inactivation (Figure 3, Appendix A). Subjection to 71 °C for 1 min led to marginal HEV inactivation only, while full HEV inactivation (over 6.0 log_10_) was reached after exposure to 71 °C for 5 min, and 80 °C for 10 min, as a shorter duration at 80 °C was not tested (Figure 3 and Appendix A).

### 3.3. Detection of Infectious HEV in Naturally Infected Pig Livers

To investigate whether the cell culture method for the detection of infectious HEV could be used for naturally infected matrices, archived pig livers were obtained that had previously been tested for HEV RNA using RT-qPCR by Wageningen Food Safety Research (WFSR) (Table 2). Three out of fourteen livers were found to be positive for infectious HEV in all tested dilutions, and one out of the fourteen livers was found positive for almost all replicates per dilution. In these four IF-positive livers, high levels of HEV RNA were observed, as indicated by the low C_q_ values obtained from the extracts. For two out of the fourteen livers (L-01 and L-11), HEV concentrations were estimated with the Bayesian MPN method, at 2.7 and 1.5 log_10_ viral particles/gram, respectively (Table 2). Of note, an enhanced number of green fluorescent cells in samples was observed in more diluted extracts as compared to less diluted ones, as was observed for liver L-07 (Appendix A), and for multiple other livers, possibly due to the simultaneous dilution of coextracted inhibitory components. Fractional positive fluorescence in dilutions, as seen for L-01 and L-11, was associated with higher C_q_ values (lower RNA levels) as compared with L-05, -06, -07 and -12, showing positive fluorescence in (almost) each dilution and low C_q_ values (approximately C_q_ 24–27). As no fractional positives were obtained for the samples with the complete absence (8/14) or presence (4/14) of infectious HEV, it was impossible to perform Bayesian MPN modelling. These samples were, therefore, estimated with 95% certainty to be below < 0.66 log_10_ viral particles/gram or beyond >2.96 log_10_ viral particles/gram, respectively. Eight of the fourteen livers did not lead to a green fluorescent signal, indicating that no infectious HEV was detected (Table 2). For the sample that served as a blank (L-13), no fluorescence was observed, nor any HEV RNA detected with RT-qPCR. In contrast, C_q_ values of L-10 (27.17 and 26.42) were indicative for relatively high HEV RNA levels, while no infectious HEV was detected with IF. For the other five out of eight IF-negative livers (L-03, 04, 08, 09 and 14), C_q_ values after freezing (all > 31, RIVM) were suggestive of low levels of HEV RNA (Table 2). For the IF-negative liver L-02, HEV RNA was detected as positive, with a C_q_ value of 35.62 by the WFSR, but not by the RIVM. This may have been due to the freezing of this specific liver subsample, as L-08 with a low C_q_ value of 35.72 prior to freezing was still detected after freezing (RIVM). Importantly, we were able to distinguish between samples with ‘intermediate’ concentrations of infectious HEV, as estimated with IF, and livers that were positive or negative for all dilutions. This difference was not fully reflected in the C_q_ values and, thus, viral RNA, however, were not due to inhibitory components in the assay. Thus, C_q_ values after freezing could serve as predictions for the detection of infectious HEV with IF. Our data showed that the successful extraction of infectious HEV was achieved in half of the naturally infected archived pig livers.

## 4. Discussion

Investigating the infectivity of HEV in pork products is of great interest to evaluate the risks for public health. Typical molecular detection methods, such as PCR-based assays, do not discriminate between infectious and noninfectious viral particles. Here, we showed a method to determine whether HEV-RNA-positive pork products contained HEV that was still infectious to host cells in cell cultures. We first optimized and implemented a cell culture method for the detection of infectious HEV [26,28]. This method allowed us to visualize the presence of replicated virus particles in infected A549/D3 cells using IF [26,28]. Earlier, when we used RT-qPCR for the detection of the replication of viral RNA in this cell culture system, we were unable to detect a consistent increase in HEV RNA in lysed infected cells and the harvested supernatant. This was likely due to the exposure of the cells to low HEV concentrations and complex matrices, like liver and sausage, or a combination of both that interfered the detection of viral replication (unpublished). In contrast, the method, as described in the present study, led to the consistent detection of infectious HEV-3c and 3e in the absence and the presence of complex matrices, such as pork products. Despite being successful, the method is currently too complex and labor intensive to be applied for the routine screening of food products. For research purposes, however, the method can be used to study the effect of food-processing steps on HEV infectivity. In the current paper, we applied the method to study the effect of different food-processing temperatures on HEV infectivity in pork products.

To study the inactivation of HEV in pork products, small spiked dried sausage pieces and spiked liver homogenates were chosen as a simulation model for naturally contaminated pig sausages and naturally infected pig liver. Herewith, standardized and controlled experimental conditions could be created, especially in light of the limited availability of highly contaminated pig sausages and livers. However, inoculating meat matrices could make virus particles more accessible to certain enzymes in dried sausage or liver homogenate, which is less likely in naturally infected livers because there the virus is located intracellularly. The intracellular location of the virus in in vivo situations necessitates extraction methods to be sufficiently rigorous to release the virus from the cells, which, however, may affect the virus inactivation rates.

Extracts from the inoculated meat matrices showed a loss of virus as compared to extracts from the culture medium, which is a normal phenomenon in food virology. It is possible that HEV was inactivated due to the composition of the meat matrices (e.g., presence of fat that shielded the virus particles or accessibility to enzymes) rather than the extraction procedure itself. HEV particles can exist as naked or as quasienveloped particles [38,39]. The presence of lipid-associated membranes of quasienveloped particles could have interfered with neutralizing antibodies and, thus, HEV infectivity in our cell culture method [39]. It is unclear whether the HEV particles were naked or quasienveloped at the time of spiking, thermal treatment and extraction, and whether this differed between the meat matrices and cell culture medium. Notably, the extraction efficiencies of the HEV-3 subtypes from the dried sausage and liver homogenate differed. It should be noted that our Bayesian MPN model compensated for the observed difference in the recovery of the virus concentrations extracted from these different matrices, and that, despite this loss of infectious HEV during sample processing, enough infectious virus remained to perform the inactivation experiments.

HEV-3c is the dominant HEV-3 subtype detected in Dutch pigs, followed by HEV-3e and HEV-3f [6,10]. Here, we investigated the effect of thermal food processing steps on HEV3c and HEV-3e inactivation in a cell culture medium and in the complex matrices of dried sausage and a liver homogenate. Similar to previously published results by Johne et al. [26], we observed that with higher temperatures, more HEV inactivation was observed in the cell culture medium. The subjection of the cell culture medium to 21 °C for 4 weeks resulted in a 3.29 log_10_ inactivation for HEV-3c/e. Similar observations were determined by Johne et al. [26] who observed a 3.2 log_10_ inactivation of HEV-3, albeit with HEV-3c specifically and with another 3c strain (47832c) [26]. The combined data from all thermal conditions tested in the cell culture medium and the two pork matrices resulted in a marginal difference between HEV-3c and HEV-3e inactivation factors. This observation provided a rationale for combining the IF data of both HEV-3 subtypes to determine the inactivation patterns for HEV-3 per matrix and not per subtype. Although the overall level of inactivation of HEV-3c was slightly more than that observed for HEV-3e, it is unclear whether this was the case for specific time–temperature combinations and/or specific matrices. Additionally, whether strain differences occurred in vivo or were specific to these robust culture(d) strains used for repetitive culture experiments remains to be determined.

Our data showed that both HEV-3c and HEV-3e extracted from the inoculated dried sausage and liver homogenate were infectious and that infectivity declined upon the prolonged subjection of HEV-3-containing matrices to different temperatures, with increasing temperatures leading to more HEV-3c/e inactivation. Notably, after incubations at 55 or 65 °C for 120 and 60 min, respectively, remaining infectious virus was still detected after extraction from these matrices. This was in agreement with the finding that pigs, inoculated with HEV-positive livers that had been heated at 56 °C for 60 min, still became infected, suggesting that HEV was not fully inactivated [23]. Furthermore, dried sausage that was inoculated with HEV and subjected to 21 °C for 4 weeks led to a substantial decrease in infectious HEV, although infectious virus remained detectable. This was interesting in light of the epidemiological association of raw pig sausages with the increased risk for HEV infection or HEV seropositivity [11,13], as these have a relatively long shelf-life. Taken together, our data could be used to estimate whether consumers of dried pig sausages are at risk for infection with HEV using quantitative microbial risk assessment strategies.

Although our simulation models provided insight into the effects of different temperatures and durations on HEV-3c/e infectivity in different pork products, they did not fully mimic the in vivo situation. Therefore, we tested our extraction and detection method on selected HEV-RNA-positive pig livers. Interestingly, we were able to demonstrate the presence of infectious HEV in half of the tested naturally infected pig livers. In the other half, no infectious HEV-3 was detected, although HEV RNA was found to be present in the majority of the samples. Whether these results reflected that no infectious HEV was present, or that it was present in such low concentrations that it could not be picked up in our assay, remains unclear. Although the C_q_ values and IF results did not fully correlate, assessing the C_q_ values prior to conducting the IF experiments may predict the likelihood of detecting infectious HEV with IF, as well as the sample dilution range needed for IF experiments and, thus, the probability of successfully estimating the viral load with Bayesian MPN methods.

Infectious HEV-3 is quite persistent, as the inactivation of HEV-3 in food matrices with reagents, such as citric and acetic acid, has been proven difficult and not fully effective [40]. A recent study on the inactivation of HEV in human milk has shown that the use of high hydrostatic pressure or the Holder pasteurization method, in which milk is subjected to 62.5 °C for 30 min, did not lead to full HEV inactivation either [41]. A possible way to further minimize food-borne HEV infection might be to expose pork or pork products to higher temperatures, longer durations of heat treatment or a combination of both. However, it should be noted that the heating of pork may, consequently, lead to differences in physicochemical characteristics of these steaks [42], which may not always be beneficial to human health. For instance, prepared fillet steaks that were well-done contained higher protein, ash, potassium and lipids than steaks that had lower doneness after preparation, and also depended on the type of preparation (pan fried versus preparation in an air fryer) [42]. The differences in physicochemical characteristics of meat products and their potential effects on human health are important to consider when proposing (heat-related) interventions for food-borne pathogens.

To summarize, we set up a method for the demonstration of infectious HEV-3 in extracts from raw pig sausage and liver using a cell culture combined with intracellular IF detection, followed by a Bayesian estimation of the infectious HEV-3 particle concentration. The method was applied to test the remaining HEV-3 infectivity in extracts of HEV-3-inoculated meat matrices after heat exposure and in naturally infected livers. The time and temperature combinations chosen were those relevant to food processing. Our findings provided a basis for future HEV infectivity experiments, not only dealing with temperature, but also other food-processing techniques, such as fermentation, acidification and high-pressure processing, that, finally, may feed into the quantitative microbial risk assessment for the consumption of food products potentially contaminated with HEV.

## Figures and Tables

**Figure 1 microorganisms-11-02451-f001:**
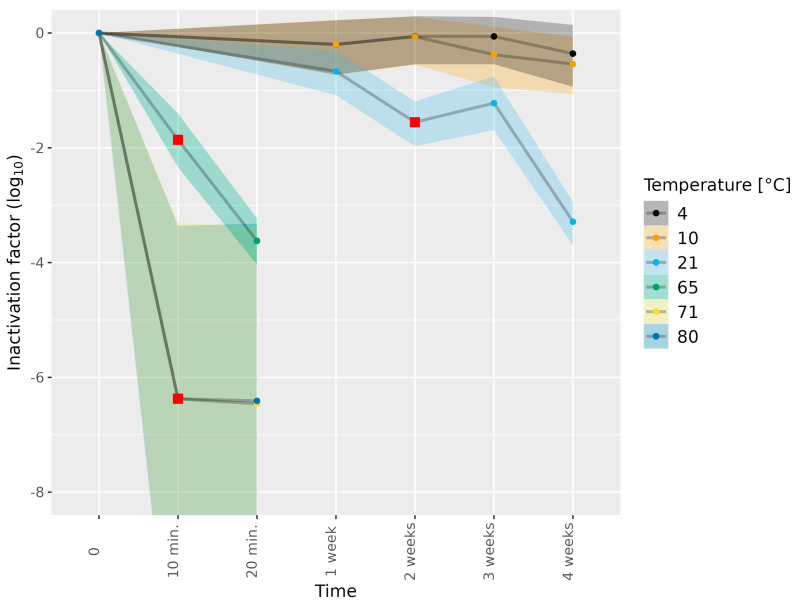
Inactivation of HEV-3c/e in cell culture medium subjected to different temperatures. HEV-3c/e inactivation factors (log_10_) were estimated using the Bayesian MPN method. Data are representative of 2 or 3 experiments performed on different days depending on the time–temperature combination (Table 1). At least four different wells of inoculated A549/D3 cells per dilution were examined. The 95% confidence intervals are indicated by colored bands. Red squares indicate the first data point, for which the 89% HDI was outside of the ROPE interval (−0.5, 0.5), and denote a meaningful reduction in infectious HEV.

**Figure 2 microorganisms-11-02451-f002:**
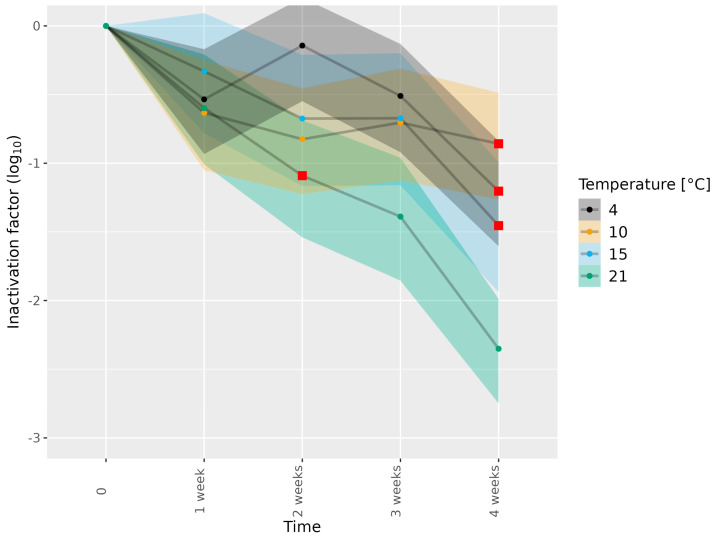
Inactivation of HEV-3c/e in dried sausage subjected to different temperatures. HEV-3c/e inactivation factors (log_10_) were estimated using the Bayesian MPN method. Data are representative of 2 or 3 experiments performed on different days depending on the time–temperature combination (Table 1). At least four different wells of inoculated A549/D3 cells per dilution were examined. The 95% confidence interval is indicated by colored bands. Red squares indicate the first data point, for which the 89% HDI was outside of the ROPE interval (−0.5, 0.5), and denote a meaningful reduction in infectious HEV.

**Figure 3 microorganisms-11-02451-f003:**
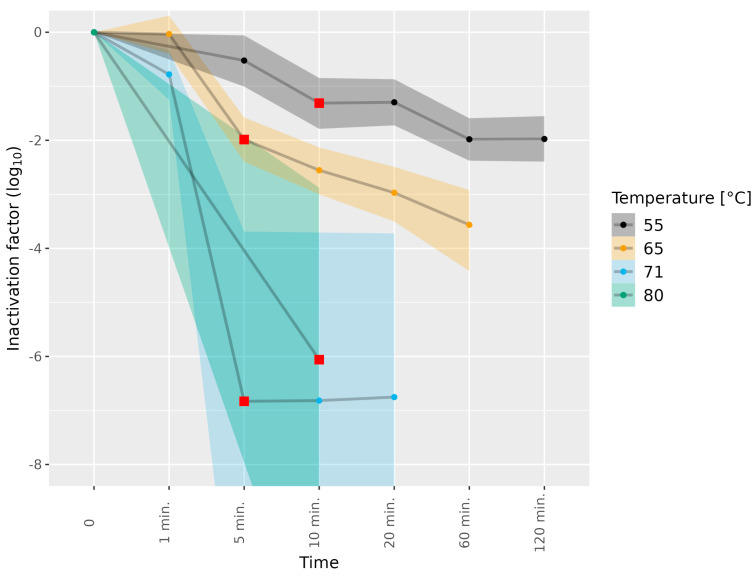
Inactivation of HEV-3c/e in liver homogenate subjected to different temperatures. HEV-3c/e inactivation factors (log_10_) were estimated using the Bayesian MPN method. Data are representative of 2 or 3 experiments performed on different days depending on the time–temperature combination (Table 1). At least four different wells of inoculated A549/D3 cells per dilution were examined. The 95% confidence interval is indicated by colored bands. Red squares indicate the first data point, for which the 89% HDI was outside of the ROPE interval (−0.5, 0.5), and denote a meaningful reduction in infectious HEV.

**Table 1 microorganisms-11-02451-t001:** Food processing conditions per matrix spiked with HEV-3c or HEV-3e.

Matrix	HEV-3 Subtype	Temperature (°C)	Duration	Number of Biological Replicates
No matrix (medium)	3c	21	1, 4 weeks	1
No matrix (medium)	3e	4, 10	1, 2, 3, 4 weeks	1
No matrix (medium)	3e	21	2, 3 weeks	1
No matrix (medium)	3e	21	1, 4 weeks	2
No matrix (medium)	3c	65	10, 20 min	1
No matrix (medium)	3e	65	10, 20 min	1
No matrix (medium)	3e	71	10, 20 min	1
No matrix (medium)	3e	80	10, 20 min	1
Dried sausage	3c	4, 10, 15, 21	1, 2, 3, 4 weeks	1
Dried sausage	3e	4, 10, 21	1, 2, 3, 4 weeks	2
Dried sausage	3e	15	1, 2, 3, 4 weeks	1
Liver homogenate	3c	55	5, 10, 20, 60, 120 min	1
Liver homogenate	3e	55	5, 10, 20, 60, 120 min	1
Liver homogenate	3c	65	1, 5, 10, 20, 60 min	1
Liver homogenate	3e	65	1, 5, 10, 20, 60 min	1
Liver homogenate	3c	71	1, 5, 10, 20 min	1
Liver homogenate	3e	71	1, 5 min	1
Liver homogenate	3e	71	10, 20 min	2
Liver homogenate	3e	80	10 min	1

**Table 2 microorganisms-11-02451-t002:** Detection of infectious HEV extracted from naturally infected pig livers. A549/D3 cells were inoculated with suspensions prepared from naturally infected pig livers. A total of 14 livers were tested. Per liver, eight wells of A549/D3 cells were inoculated per dilution (5×, 10× or 100×). The number of positive wells per dilution is indicated between brackets. Log_10_ viral particles/gram were estimated using the Bayesian MPN model, with a 95% confidence interval (C.I.). Average C_q_ obtained with RT-qPCR values are given for the viral extract used for cell culture (RIVM) and testing in monitoring studies for the presence of HEV RNA (WFSR). Dilutions and C_q_ values labelled as n.d. indicate that fluorescence and HEV RNA, respectively, were not detected.

Pig Liver ID	Dilution with Positive IF Results	Log_10_ Viral Particles/g (95% C.I.)	C_q_ Values (RIVM)	C_q_ Values (WFSR)
L-01	5× (8)/10× (3)/100× (1)	2.7 [2.42, 2.96]	31.38	33.75
L-02	n.d.	<0.66	n.d.	35.62
L-03	n.d.	<0.66	31.24	32.17
L-04	n.d.	<0.66	31.09	26.53
L-05	100× (8)	>2.96	25.68	26.53
L-06	100× (8)	>2.96	24.58	24.17
L-07	100× (8)	>2.96	25.52	27.24
L-08	n.d.	<0.66	35.71	35.72
L-09	n.d.	<0.66	32.88	33.25
L-10	n.d.	<0.66	27.17	26.42
L-11	5× (0)/10× (2)/100× (0)	1.5 [0.66, 2.09]	31.18	29.55
L-12	5× (8)/10× (8)/100× (7)	>2.96	27.44	22.15
L-13	n.d.	<0.66	n.d.	n.d.
L-14	n.d.	<0.66	34.66	34.26

## Data Availability

The data presented in this study are available in this article and Appendix A.

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
