# Peer review of "Thermal Inactivation of Hepatitis E Virus in Pork Products Estimated with a Semiquantitative Infectivity Assay"

_microorganisms, 2023, doi:10.3390/microorganisms11102451_

Round 1
Reviewer 1 Report
In the present manuscript, Stunnenberg et al. have investigating thermal inactivation of HEV in different pork products using a cell culture-based assay that allows to detect the presence of infectious particles. This paper is of high interest for the field as a method that allows quantification of HEV infectious particles are not commonly available and difficult to set up. Few studies have addressed thermal inactivation of HEV in pork products and it is critical to better understand how meat products need to be processed before consumption to better control zoonotic transmission of the virus. The manuscript is clear and well written. Here are a few points for considerations.
-
Line 61-75/Figure S1: This figure is interesting as it gives a summary of all the temperatures and conditions tested. I was just wondering if there were some contradictory results between these studies.
-
Line 106: what do you mean by “an unknown titer”?
-
Virus stocks were produced from supernatant and cell lysates. Have you checked whether these stocks that were used for the spiking/infections contain naked and/or quasi-enveloped viruses? Could this influence thermal inactivation? This point could be discussed in the discussion if relevant.
-
Lie 382-391/line 396-399 (89% HDI) and Figure S4: I find the text and figure difficult to understand. Could this be clarified/simplified?
-
Line 437-439: I find this sentence difficult to follow
-
Line 498: have you check that the liver sample did not contain inhibitors that could have influenced RT-qPCR and Cq values?
-
Line 518-520: Any hypothesis why an increase in HEV RNA was not observed?
-
Have you tried to assess thermal inactivation of the viruses in naturally infected pig livers to ensure that your spiking protocol is suitable and confirm your results?
English language is correct. The text is clear and the paper is well-written.
Author Response
Dear reviewer,
On behalf of all co-authors, I would like to take this opportunity to thank you for constructive comments on our manuscript. Good suggestions have been provided to improve the quality of the manuscript. We revised the manuscript and have addressed all your reviewer comments and suggestions. In the document attached, we detail our point-by-point responses to your comments and suggestions. For better readability we highlighted our responses in red.
With kind regards, Saskia Rutjes

Reviewer 2 Report
Thank you for interesting article.
I have some remarks to do, therefore there are some parts that need revision.
Introduction:
Lane 38: please, add this sentence with relevant updated reference: “and also in different breeding systems”. (Ianiro, G., Pavoni, E., Aprea, G., Romantini, R., Alborali, G. L., D'Angelantonio, D., ... & Di Bartolo, I. (2023). Cross-sectional study of hepatitis E virus (HEV) circulation in Italian pig farms. Frontiers in Veterinary Science, 10, 1136225).
Materials and Methods:
English and writing style are difficult to follow in this section, please check grammar.
Lanes 473, 481 and 492: please, replace “expression” with “amount”
In the experimental studies, the authors should report the number of experimental repetitions carried out for each time point in the respective sections in MeM; standard deviations should be placed into the graphs in figures 1, 2 and 3.
The captions of the figures must not include the description of the method used. Please, report methods used in MeM section.
Results:
Lane 366: this sentence must be placed in the discussions
Materials and Methods:
English and writing style are difficult to follow in this section, please check grammar.
Author Response

(The authors gave the same response as above.)

Round 2
Reviewer 2 Report
Please, address the following minor editings:
Line 231: please, remove the expression “was performed”.
Figures 1, 2 and 3: The following sentence to be placed in MeM - “Data are representative of 2 or 3 experiments performed on different days depending on the time/temperature combination (Table 1).”
Caption of Table 2: The following sentence to be placed in MeM - “A549/D3 cells were inoculated with suspensions prepared from naturally-infected pig livers. A total of 14 livers were tested. Per liver, eight wells of A549/D3 cells were inoculated per dilution (5×, 10× or 100×)”
Author Response
Dear reviewer,
Thanks for the minor suggestions to further improve the manuscript. Please find below our point-by-point responses to the comments that we included in the revised version of the manuscript. and suggestions of the three reviewers. For better readability we highlighted our responses in italic, line numbers refer to the revised document that includes track changes.
Please, address the following minor editings:
Line 231: please, remove the expression “was performed”.
The expression ‘was performed’ was removed accordingly (line 232)
Figures 1, 2 and 3: The following sentence to be placed in MeM - “Data are representative of 2 or 3 experiments performed on different days depending on the time/temperature combination (Table 1).”
We included the suggested sentence in the MeM (line 206 – 208)
Caption of Table 2: The following sentence to be placed in MeM - “A549/D3 cells were inoculated with suspensions prepared from naturally-infected pig livers. A total of 14 livers were tested. Per liver, eight wells of A549/D3 cells were inoculated per dilution (5×, 10× or 100×)”
We thank the reviewer for this suggestion for clarification, we included the sentence concerned in MeM, lines 334 – 336.
We hope that by this revision based on your comments the revised document will be now acceptable for publication in the special issue of Microorganisms.
With kind regards,
Saskia Rutjes
